# Adsorptive Removal of Reactive Yellow 145 Dye from Textile Industry Effluent Using *Teff* Straw Activated Carbon: Optimization Using Central Composite Design

Melkamu Kifetew [1], Esayas Alemayehu [2,3,*], Jemal Fito [4], Zemene Worku [1], Sundramurthy Venkatesa Prabhu [5] and Bernd Lennartz [6,*]

1. Department of Environmental Engineering, College of Biological and Chemical Engineering, Addis Ababa Science and Technology University, Addis Ababa P.O. Box 16417, Ethiopia; melkam1703@gmail.com (M.K.); hezeni@gmail.com (Z.W.)
2. Faculty of Civil and Environmental Engineering, Jimma University, Jimma P.O. Box 378, Ethiopia
3. Africa Center of Excellence for Water Management, Addis Ababa University, Addis Ababa P.O. Box 1176, Ethiopia
4. Institute for Nanotechnology and Water Sustainability (iNanoWS), College of Science Engineering and Technology, University of South Africa, Florida Science Campus, Johannesburg 1710, South Africa; fitojemal120@gmail.com
5. Department of Chemical Engineering, College of Biological and Chemical Engineering, Addis Ababa Science and Technology University, Addis Ababa P.O. Box 16417, Ethiopia; haiitsvp@gmail.com
6. Faculty of Agricultural and Environmental Sciences, University of Rostock, Justus-Von-Liebig-Weg 6, 18059 Rostock, Germany
* Correspondence: esayas16@yahoo.com (E.A.); bernd.lennartz@uni-rostock.de (B.L.)

**Abstract:** This study aimed to optimize the preparation condition of activated carbon using *Teff* straw as a precursor material via evaluating its potential in terms of maximizing the adsorptive removal of Reactive Yellow 145 dye (RY 145) from aqueous solutions. Selective factors, such as activation time, activation temperature, and impregnation ratio on the preparation of *Teff* Straw-based Activated Carbon (TSAC) were optimized using response surface methodology (RSM). A quadratic regression model with estimated coefficients was developed by RSM and it was observed that model predictions were matched with experimental value with an acceptable $R^2$ value (0.98). Further, the TSAC prepared at optimal condition was characterized using Fourier-transform infrared (FTIR), Scanning electron microscopy (SEM), and X-ray diffraction (XRD) techniques. The TSAC prepared at optimal condition showed anionic nature with a BET surface area of 627.7 m$^2$/g. In addition, important adsorptive parameters (contact time, solution pH, adsorbent dose, and dye concentration) were evaluated through batch experiments. In such a way, it was determined that 2 h for activation time, 539 °C for activation temperature, and impregnation ratio of 5 g of phosphoric acid per 1 g of TSAC were optimal for efficient adsorption with maximum removal of 98.53% for RY 145 dye. In addition, the TSAC was subjected to test in order to determine its adsorptive performance by treating real textile industry effluent for examining its Chemical Oxygen Demand (COD) removal potential. The results showed that 76% COD was removed from the real textile effluent, which met Ethiopian Environmental Protection Authority (EPA) standard. The finding of this paper asserts that this material is a good and low-cost bio-sorbent that can be used for the removal of pollutants from textile wastewater. Nevertheless, additional investigations of the adsorbents including regeneration options are advisable to draw explicit conclusions.

**Keywords:** bio-sorbent; dyes removal; response surface methodology; *Teff* straw; textile industry effluent

## 1. Introduction

Dyes have a variety of industrial applications such as in paper, pulp, textiles, plastics and leather industries, food and beverage companies, pharmaceuticals, and paint manufacturing [1]. Due to process inefficiency, waste streams carry 10–15% of dyes that are used

during processing [2]. Based on estimates, the textile industry has been ranked second after agriculture in terms of pollution. The industry uses more than 8000 chemicals during various processes [3]. In addition, textile industry is water intensive—about 200 L of water is used to produce 1 kg of textile; hence, the amount of wastewater generation seems to be huge. In this line, the discharge of inadequately treated dye stuff effluent into water bodies is undesirable and results in pollution for receiving water bodies, causing health-related problems [4]. Accordingly, the wastewaters from textile effluents should be treated properly for recycling [5].

Textile effluents were reported to contain toxic chemicals that produce high chemical oxygen demand (COD), biological oxygen demand (BOD), toxic heavy metals, dissolved solids (DS), suspended solids (SS) and non-biodegradable materials [3,6]. Living organisms are significantly impacted by these toxic chemicals that result in increasing risks of health issues [5]. Furthermore, their presence in water bodies inhibits the penetration of sunlight that adversely affects photosynthesis process. This affects air–water interface on oxygen transfer, which is most essential for self-purification of aquatic system and healthy life of aquatic ecosystem. Textile contaminants are toxic to water organisms and exhibit a significant uproar on food chain. Presence of textile effluent in soil also results in clogging of soil's pore that can reduce productivity [3]. In consequence, textile wastewater has to be treated appropriately before being released to environment [3,7]. In this context, there are different methods that are available for removing the pollutants. Among them, photo catalytic degradation [8], membrane separation [9], microbial degradation [10], chemical oxidation [11] are the most effective; however, these methods have disadvantages of being expensive with high investment cost, regular maintenance, complicated operation and fouling issues [8,12].

When compared to the different wastewater treatment techniques, adsorption is simple and easy to operate, flexible, and highly efficient, with a relatively low cost [13]. However, adsorption using commercial activated carbon is limited due to its high cost and regeneration problem [14]. In recent time, to overcome the challenge, investigation into agricultural and industrial residue as feedstock for preparation of activated carbon has become the focus of attention. Due to their availability in natural abundance, their relative low cost, and the fact that they are ecologically friendly [15], biomass sources are considered as potential feedstock. Many researchers have investigated various low-cost adsorbents from agricultural waste including *Teff* straw (TS) and *Teff* husk [16,17], coffee husk [18], cocoa shell [19], rape stalk [20], hazelnut and almond shell [21], seed of *Capsicuum annuum* [22], waste from citrus [23], and rice husk [24]. Among these, *Teff* straw is receiving more attention because of its abundance, local availability, and high binding affinity to potentially hazardous contaminants.

*Teff* is known to be a popular indigenous crop in Ethiopia. Generally, in Ethiopia, a significant amount of *Teff* hay can be generated as agricultural residue obtained from post-harvest processing. Hence, plenty of *Teff* hay can be obtained in a relatively inexpensive manner. In this line, *Teff* straw (TS), which is the dry stalks part of *Teff* hay, was characteristically proven to be composed of 28.99% hemicellulose; 37.1% cellulose; 17.85% lignin; and 8.55% extractives by weight [16,17]. Previous researches on TS determined it to be promising feed stock for preparing low-cost bio-sorbent to remove Cr (VI) [7,25,26]; Pb(II) [27]; and phenol red dye [16]. Nevertheless, TS-based activated carbon, hereinafter abbreviated as TSAC, has not been tested for the treatment of textile industry wastewater. Azo dyes are extensively used coloring agent in textile industries [28]. In addition, they generate a dye-containing wastewater that has significant toxicity, threatens all kinds of life, and exhibits substantial impacts on human being's natural health. One of the important azo dyes, Reactive Yellow 145 dye, was considered in this study. Therefore, the aim of the current study was to (i) optimize the preparation condition (activation time, activation temperature, and impregnation ratio) for TSAC using response surface methodology (RSM), (ii) characterize the TSAC which was prepared at optimal condition using Fourier-transform infrared (FTIR), Scanning electron microscopy (SEM), and X-ray diffraction (XRD) techniques, (iii) evaluate the adsorption potential of TSAC

on the adsorptive removal of Reactive Yellow 145 dye (RY 145) at different pH value, initial concentration of dye, adsorbent dose, and contact time via one variable at a time method, and (iv) examine the adsorptive removal of COD from real textile industry effluent using the optimized TSAC.

## 2. Materials and Methods

### 2.1. Preparation of Activated Carbon

Activated carbon was prepared using *Teff* straw (TS), which is one of the most predominant agro-residual wastes. In the present study, TS was obtained from the farmers surrounding to Addis Ababa, Ethiopia. Before commencing the chemical activation of TS, impurities were removed by washing several times, using distilled water. Further, it was dried using a hot air oven (UN30, India) for 24 h at 105 °C. The dried TS was ground and sieved to screen particle size of 300 μm. The TS produced was activated by using one mole per liter (1 M) phosphoric acid (85%) for 24 h. The mixtures were stirred well to ensure appropriate mixing. The influence of impregnation ratio on the removal capacity was investigated in the impregnation ratio of 1:3, 1:45, and 1:6 (weight of TS: weight of phosphoric acid). The impregnated samples were dried in air under the sun. Consequently, they were activated in a muffle furnace (Nabertherm GmbH, Lilienthal, Germany) in nitrogen atmosphere with heating at 400, 500, and 600 °C for 1, 2, and 3 h at 10 °C/min with 180 cm$^3$/min of flow rate. After the activation process, the samples were subjected to cooling under nitrogen atmosphere. The prepared TSAC was washed using distilled water until pH reached constant value. Further, heating commenced in a hot air oven at 105 °C for 12 h. The acquired TSAC products were stored in an airtight container for the optimization experimental studies (Figure 1).

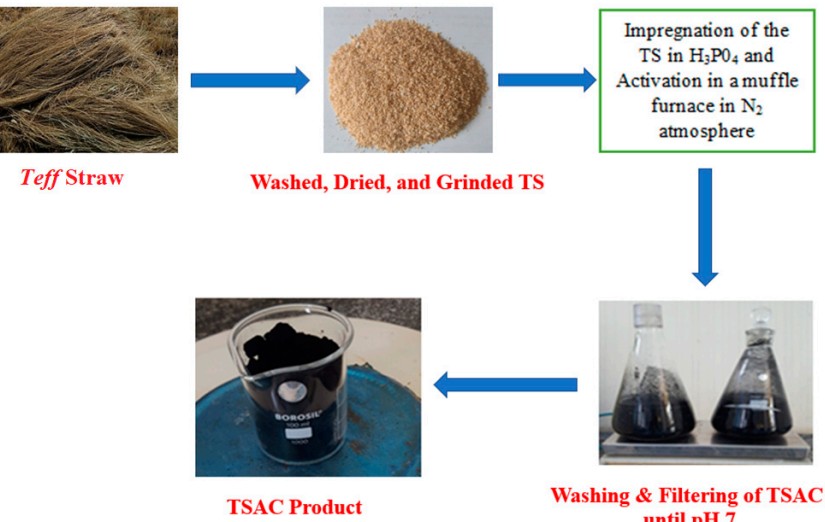

**Figure 1.** Schematic diagram of TSAC preparation.

### 2.2. Preparation of Reactive Yellow 145 Dye-Simulated Industrial Wastewater

Reactive yellow 145 dye (RY 145) was obtained from Yirgalem Addis Textile Factory, Ethiopia. RY 145 dye is used widely in dying processes in textile industries and thus it can be found in industrial wastewater. The general characteristics and chemical structure of the reactive dye are presented in Table 1 and Figure 2, respectively. The desired experimental solution was obtained by diluting the predetermined stock solution as per the desired concentration. Adsorptive efficiency of the TSAC was evaluated through batch experimental runs according to predetermined Design of Experiment (DOE; Stat-Ease, Design-Expert 13). The removal efficiency was determined using UV spectrophotometer (V-770, Easton, MD, USA).

**Table 1.** Characteristics of RY 145 Dye145 [29,30].

| Characteristics | RY 145 Dye |
|---|---|
| C.I. generic name | C.I. RY 145 dye |
| Molecular formula | $C_{28}H_{20}ClN_9Na_4O_{16}S_5$ |
| Molecular weight | 1026.25 |
| $\lambda_{max}$ | 418 nm |

Note: Where, C.I.: Color index.

**Figure 2.** Chemical Structure of RY 145 [28–30].

*2.3. Textile Effluent Collection and Chracterization*

The textile wastewater effluent used in this investigation was obtained from *Yirgalem Addis Textile Factory, Ethiopia*. A sample from textile wastewater was obtained using grab sampling method. To avoid any unwanted characteristic changes in collected wastewater, the effluent was preserved at 4 °C. Further, the wastewater effluent was subjected to characterization by adopting standard procedures (Table 2). The TSAC was evaluated for its chemical oxygen demand (COD) removal efficiency from the textile industry wastewater.

**Table 2.** Selective characteristics of the raw textile effluent.

| Temp (°C) | PH | COD (mg/L) | EC (μS/cm) | TDS (mg/L) |
|---|---|---|---|---|
| 10.28 | 9.8 | 804 | 7673 | 3321 |

*2.4. Design of Experiments towards Optimizing the Condition for TSAC Preparation*

In order to optimize the preparation condition, adsorption experiments were carried out using the TSAC via 20 sets of 250 mL Erlenmeyer flasks. Accordingly, a solution of 100mg/L of the RY 145 dye was placed in each flask with 0.3 g of the TSAC. Adsorption studies were conducted at isothermal shaker at 120 rpm to reach equilibrium. After the adsorption process, the concentration of the RY 145 dye in the supernatant was analyzed at 418 nm using UV spectrophotometer (V-770, Easton, MD, USA). Dye removal efficiency was determined through percent decolonization using Equation (1) [31].

$$\text{Removal}\ (\%) = \frac{(Co - Ce)}{Co} \times 100, \tag{1}$$

where Ce and Co refer to the equilibrium concentration of dye and initial concentration (mg/L) of the dye, respectively.

In the present study, a Central Composite Design (CCD) was executed through response surface methodology (RSM) to optimize the activation parameters for preparing the activated carbon. In this context, using the RSM technique, quantitative experimental data were applied to determine a model equation; further, the optimal condition was determined for the adsorption process [32]. In such a way, aiming to optimize the process, three important key factors were selected, namely the amount of phosphoric acid (g) per one g of TASC, activation time, and activation temperature. The percent removal of dye was considered as the response. Ranges and levels of the selected independent variables are presented in Table 3.

**Table 3.** Levels and ranges of the independent variables used for the design of experiment.

| Independent Variables | Range and Levels | | |
|---|---|---|---|
| | −1 | 0 | 1 |
| Heating time (h) | 1 | 2 | 3 |
| Temperature (°C) | 400 | 500 | 600 |
| Impregnation ratio | 3 | 4.5 | 6 |

2.4.1. Regression Model

An empirical model (Equation (2)) that correlates the activation process variables for preparing TSAC to the dye removal was developed [33].

$$Y = b_0 + \sum b_i x_i + \sum b_{ii} x_i{}^2 + \sum b_{ij} x_i x_j, \tag{2}$$

where Y is predicted response; $b_0$ is the constant; $b_i$ is linear coefficient; $b_{ii}$ is quadratic coefficient; $b_{ij}$ refers the interaction regression polynomial coefficients; $x_i$ and $x_j$ are the values of the selected independent factor, respectively.

2.4.2. Statistical Optimization and Interaction Effect

Based on the experimental response data generated using the CCD combination, the quadratic model was developed using RSM to predict the optimum preparation condition. The design expert software (Stat-Ease, Design-Expert 13) was used for optimizing the response surface model to achieve the maximized RY 145 dye removal. Further, the validity of the model was tasted using the analysis of variance (ANOVA). Additionally, the 3D surface plots were used to analyze the interaction effects of the variables on the response [34].

2.4.3. Model Validation

The RSM model which correlates the dye removal efficiency and selected parameters was predicted compared with the results obtained from the batch adsorptive experiments to evaluate the model validity.

*2.5. Characterization of the Optimal TSAC*

To determine the available surface functional groups on the optimized TSAC, Fourier transform analysis (FTIR) was carried out (Perkin Elmer, Annapolis, MD, USA). Surface texture study of the activated carbon was examined using scanning electron microscopy (SEM) (INSPECT, F50, USA). The crystalline phase of TSAC was characterized using the XRD instrumental technique (XRD 7000, Tokyo, Japan). Surface area determination was performed using Brannuer–Emmett–Teller (BET) (SA-9600 Series, Tokyo, Japan). Point of zero charge was determined using the pH drift method.

*2.6. Yield of Activated Carbon Prepared under the Optimized Condition*

The yield of the activated carbon prepared from *Teff* straw is known to be the ratio of the mass of the activated carbon obtained to the mass of the precursor. It was determined based on Equation (3) [15]:

$$\text{Yield of TSAC (\%)} = \frac{m(g)}{m_o(g)} \times 100, \tag{3}$$

where $m(g)$ is the acquired mass of the TSAC and $m_o(g)$ is the mass of the precursor used, in this case, TS.

*2.7. Evaluation of the Process Variables for RY 145 Adsorption*

In order to evaluate the processes of RY 145 adsorption through variations in the solution pH, initial concentration, adsorbent dose, and contact time, one-variable-at-a-time

approach was applied. Consequently, the influence of different adsorbate concentration of 0.1, 0.2, 0.3, 0.4 and 0.5 g/L; pH of the solution of 2, 4, 6, 8, and 10; adsorbent dose of 0.1, 0.2, 0.3, 0.4, 0.5 g; and contact time of 30, 60, 90, 120, 150 min were evaluated. The variable range selection was based on the literature survey. While studying the influence of the pH value, the solution was adjusted using 0.1 M NaOH and HCl.

### 2.8. Adsorptive Removal of COD from Real Textile Industry Effluent Using the Optimized TSAC

COD removal potential of the prepared TSAC was determined by adding 0.3 g of the activated carbon into 100 mL of the textile effluent. The COD value for the effluent before and after the treatment was determined at 620 nm by using spectrophotometry method (HI 83099 COD and Multipara meter photometer, Hanna instruments, Bang Poo, Thailand). COD removal efficiency was determined using Equation (4).

$$\text{Removal of COD } (\%) = \frac{(COD_0 - COD_f)}{COD_0} \times 100, \tag{4}$$

where $COD_o$ and $COD_f$ refer to the initial and final values of COD, respectively.

## 3. Results and Discussions

### 3.1. Central Composite Design

The design combinations obtained from CCD using Design Expert 13 software were applied on the independent selected parameters. In this research work, three factors and three level CCD were used to evaluate and optimize the TSAC preparation process variables on the responses such as RY 145 dye removal efficiency of simulated industrial effluent. The selected processing factor and their levels are presented in Table 3. As seen in the table, a total number of 20 experimental runs was required for executing the RSM. Table 4 illustrates the design matrix of parameters and response in terms of % removal of RY 145.

**Table 4.** The CCD design matrix and their corresponding response using TSAC.

| Run | Activation Temperature, (°C) | Activation Time (h) | Amount of $H_3PO_4$ (g) | RY 145 Dye Removal Efficiency (%) Experimental | RY 145 Dye Removal Efficiency (%) Predicted |
|-----|------------------------------|---------------------|-------------------------|-------------|-----------|
| 1 | 400 | 1 | 3 | 46.14 | 46.9322 |
| 2 | 600 | 1 | 3 | 50.75 | 48.9327 |
| 3 | 400 | 3 | 3 | 44.05 | 41.9205 |
| 4 | 600 | 3 | 3 | 47.5 | 48.2717 |
| 5 | 400 | 1 | 6 | 54.6 | 53.0979 |
| 6 | 600 | 1 | 6 | 76.56 | 77.9591 |
| 7 | 400 | 3 | 6 | 52.34 | 53.4254 |
| 8 | 600 | 3 | 6 | 84.16 | 82.6374 |
| 9 | 400 | 2 | 4.5 | 72.32 | 74.0738 |
| 10 | 600 | 2 | 4.5 | 88.51 | 89.6800 |
| 11 | 500 | 1 | 4.5 | 82.95 | 84.0800 |
| 12 | 500 | 3 | 4.5 | 82.12 | 83.9133 |
| 13 | 500 | 2 | 3 | 69.23 | 71.6128 |
| 14 | 500 | 2 | 6 | 91.34 | 91.8785 |
| 15 | 500 | 2 | 4.5 | 97.84 | 95.4860 |
| 16 | 500 | 2 | 4.5 | 94.98 | 95.4860 |
| 17 | 500 | 2 | 4.5 | 95.24 | 95.4860 |
| 18 | 500 | 2 | 4.5 | 97.41 | 95.4860 |
| 19 | 500 | 2 | 4.5 | 95.51 | 95.4860 |
| 20 | 500 | 2 | 4.5 | 97.78 | 95.4860 |

### 3.1.1. Fitting Model Equation and Statistical Analysis

According to the design matrix, adsorption investigations were carried out to examine the effect of activation temperature (°C), activation time (h), and impregnation ratio on the percentage RY 145 dye removal efficiency using TSAC bio-sorbent for the simulated wastewater. The results on the removal percentage of RY 145 by TSAC were determined and are recorded as listed in Table 4. Using ANOVA, different models were tested via sequential model sum of squares. Additionally, summary of model statistics were analyzed to check the adequacy and appropriateness of various model that represents percentage RY 145 removal efficiency by TSAC bio-sorbent. Results of these tests are given in Tables 5 and 6, for adsorptive removal of RY 145. It seen in Table 5 that the cubic model was determined to be aliased. As per Table 5, the quadratic model was determined to have maximum "Adjusted $R^2$" and "Predicted $R^2$" values; therefore, quadratic model was determined to be appropriate for further analysis.

**Table 5.** Model Summary Statistics.

| Source | Std. Dev. | $R^2$ | Adjusted $R^2$ | Predicted $R^2$ | PRESS | |
|---|---|---|---|---|---|---|
| Linear | 19.15 | 0.2180 | 0.0713 | −0.4100 | 10,581.55 | |
| 2FI | 20.72 | 0.2559 | −0.0875 | −4.5445 | 41,608.84 | |
| Quadratic | 2.16 | 0.9938 | 0.9882 | 0.9376 | 468.64 | Suggested |
| Cubic | 2.19 | 0.9962 | 0.9879 | −2.2063 | 24,062.15 | Aliased |
| Coefficient of variance = 2.83 | | | Adequate precision = 55.767 | | | |

**Table 6.** ANOVA of the second-order polynomial equation for RY 145 dye removal.

| Source | Sum of Squares | df | Mean Square | F-Value | *p*-Value | |
|---|---|---|---|---|---|---|
| Model | 7458.05 | 9 | 828.67 | 178.22 | <0.0001 | significant |
| A—Activation temperature | 608.87 | 1 | 608.87 | 130.94 | <0.0001 | |
| B—Activation time | 0.0689 | 1 | 0.0689 | 0.0148 | 0.9055 | |
| C—Amount of $H_3PO_4$ | 1026.78 | 1 | 1026.78 | 220.82 | <0.0001 | |
| AB | 9.46 | 1 | 9.46 | 2.03 | 0.1842 | |
| AC | 261.29 | 1 | 261.29 | 56.19 | <0.0001 | |
| BC | 14.26 | 1 | 14.26 | 3.07 | 0.1105 | |
| $A^2$ | 509.35 | 1 | 509.35 | 109.54 | <0.0001 | |
| $B^2$ | 363.03 | 1 | 363.03 | 78.07 | <0.0001 | |
| $C^2$ | 519.13 | 1 | 519.13 | 111.65 | <0.0001 | |
| Residual | 46.50 | 10 | 4.65 | | | |
| Lack of Fit | 37.37 | 5 | 7.47 | 4.09 | 0.0741 | not significant |
| Pure Error | 9.13 | 5 | 1.83 | | | |
| Cor Total | 7504.55 | 19 | | | | |

### 3.1.2. Evaluation of Results from Design of Experiments

A three-factor and three-level CCD coupled with RSM was carried out for optimizing three chosen independent process factors, such as activation temperature (A), activation time (B), and impregnation ratio (C) on the percentage RY 145 dye removal efficiency. From the RSM analysis, a second polynomial model of equation that describes the percentage of RY 145 removal efficiency can be expressed as follows (Equation (5)):

$$\text{RY 145 removal, (\%)} = 95.486 + 7.803A + 10.133C - 13.61A^2 - 11.49B^2 - 13.74C^2 + 5.715AC. \tag{5}$$

In the equation, positive signs in front of the terms show synergistic effect while negative signs show antagonistic effect. The standard deviation for Equation (5) was determined to be 2.15591. This shows that the predicted value for the response is more accurate and closer to the actual values.

### 3.1.3. Adequacy of the Obtained RSM Model

Analysis of variance (ANOVA) statistical approach was carried out to analyze adequacy of obtained model and its significance. The results are presented in Table 6. From the ANOVA analysis, the model f-value for percentage RY 145 removal was 178.22, and the value indicated that the developed model determined to be highly adequate and significant. Furthermore, the determination coefficient ($R^2$) of the model was 0.9938 (Table 5), which indicated that the predicted results from the developed polynomial model can be excellent. Moreover, the F-values for lack of fit of 37.37 for RY 145 removal suggested that the model was determined to be well-fitted with the experimental data. In this context, there is only a 0.005% RY 145 removal chance for a lack of fit F-value, which could occur due to noise. From Table 6, it was also observed that *p*-value and F-value of each variable was significant for model except, B, AB interaction, and BC interaction. In addition, low value of the coefficient of variation (CV, 2.83%) cleared that the variation was acceptable and satisfactory. Furthermore, all regression coefficient values were found to be fitted with the significance levels as presented in Table 6.

Figure 3 presents the experimental versus predicted values of RY 145 removal efficiencies. It was observed that all values approximately fitted to 45° line, which demonstrated that the developed RSM model was well-appropriate to fit dye removal data. Hence, the results illustrated the accuracy and applicability of the CCD model for RY 145 dye removal process optimization.

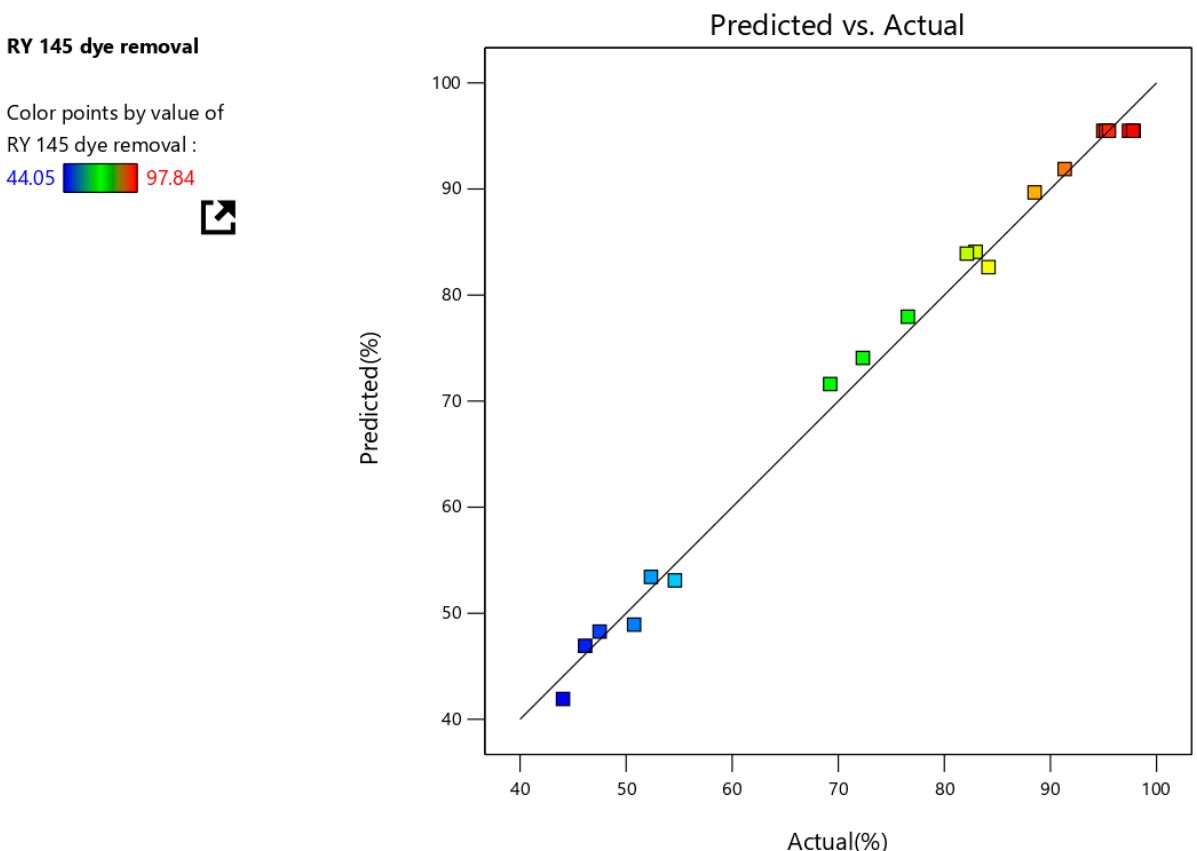

**Figure 3.** The graphical illustration for the actual and predicted dye removal efficiency.

### 3.1.4. The Effect of Operating Parameters on the Removal of RY 145 Dye

Table 4 shows that the RY 145 dye adsorptive removal efficiency of the TSAC varied between 44.05 and 97.84%. These variations in removal efficiency confirmed the significant influence of independent factors on the adsorption process which predict the quality of the synthesized product of activated carbon [1]. The results indicated that the amount

of phosphoric acid ($H_3PO_4$) and activation temperature had significant effect on the dye removal, whereas activation time had less effect. Further, the quadratic effect of the amount of $H_3PO_4$ on the removal of the dye was higher than that of activation temperature and activation time. Removal efficiency was maximized with high amount of $H_3PO_4$ and temperature while heating time had low effect. With an increase in amount of $H_3PO_4$ and temperature, porosity of the activated carbon can be increased due to the formation of micro pores and gasification reaction associated with $H_3PO_4$ activation. This was due to the fact that the pores opened and became wide, which resulted in increase in porosity and consequently the increase in RY 145 dye removal efficiency [5,35,36].

### 3.2. Effect of Operating Parameters on the Removal Efficiency of RY 145 Dye

In general, 3D response surface plots and corresponding counter-plots are developed using RSM model to investigate the individual and interactive effect among the selected variables on the responses. In the present study, the optimal value for each selected factor for maximizing the dye removal efficiency of RY 145 dye was also determined.

### 3.2.1. Interaction Effect between Activation Temperature and Heating Time

The selected parameters, activation temperature and heating time on RY 145 dye removal efficiency, were investigated by TSAC bio-sorbent, and the results are shown in Figure 4a,b. Figure 4a,b shows that with increase in activation temperature and heating time while keeping the amount of $H_3PO_4$ as constant, the dye removal efficiency also increased until a certain limit. With further increase in the temperature, dye removal efficiency had showed a decreasing trend. Such a declined response could have occurred due to collapse and shrinkage of the carbon framework caused by excess temperature [37]. This result is in agreement with the report documented by [38], where the activated carbon was prepared using cassava peel, which was chemically activated by KOH. In the present study, while studying the interaction effect, an increase in activation temperature from 400 °C to 527.98 °C had showed a good positive influence that can increase the removal efficiency of TSAC. After 527.98 °C, it was observed that the response had exhibited a decreasing pattern. In such a way, from the prediction, the analysis showed that 1.998 h of activation time can produce effective TSAC that have maximum dye removal efficiency (96.60%).

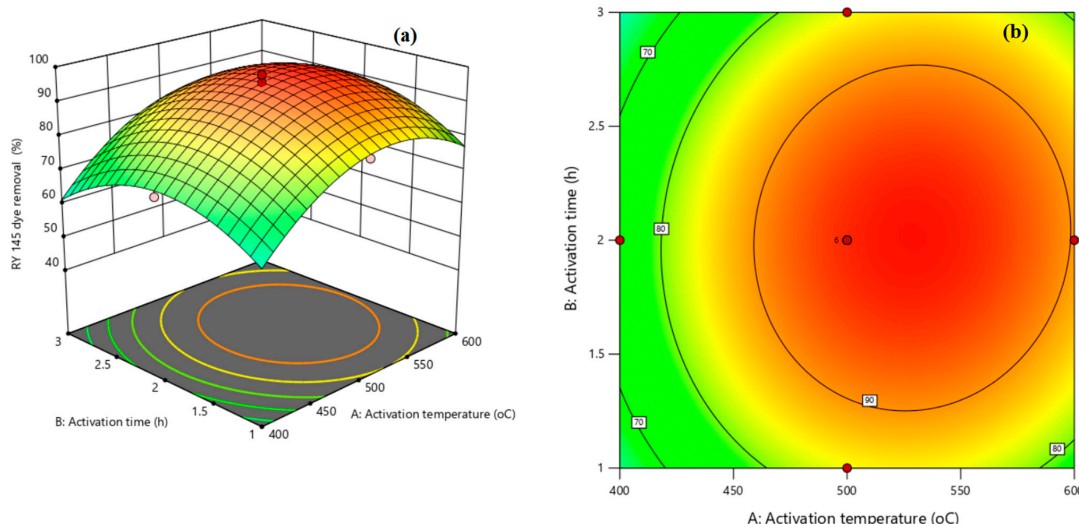

**Figure 4.** Response surface plots for the effects of activation temperature and heating time on the 3D response (**a**) and its corresponding counter plot (**b**).

### 3.2.2. Interaction Effect between $H_3PO_4$ and Activation Temperature

*The 3D response surface* prediction and its corresponding counter plot for the interaction effect of $H_3PO_4$ and activation temperature are illustrated in Figure 5a,b. In the

present study, an increase in amount of $H_3PO_4$ from 3.0 to 5.177 g showed a good positive influence, which increased dye removal efficiency. Beyond this limit, further increase in $H_3PO_4$ exhibited a decrease in response (Figure 5a,b). Hence, the prediction from the interaction revealed that the optimal values that resulted in the maximum dye removal effeicency (99.24%) for amount of $H_3PO_4$ and activation temperature were determined to be 5.177 g and 538.03 °C, respectively.

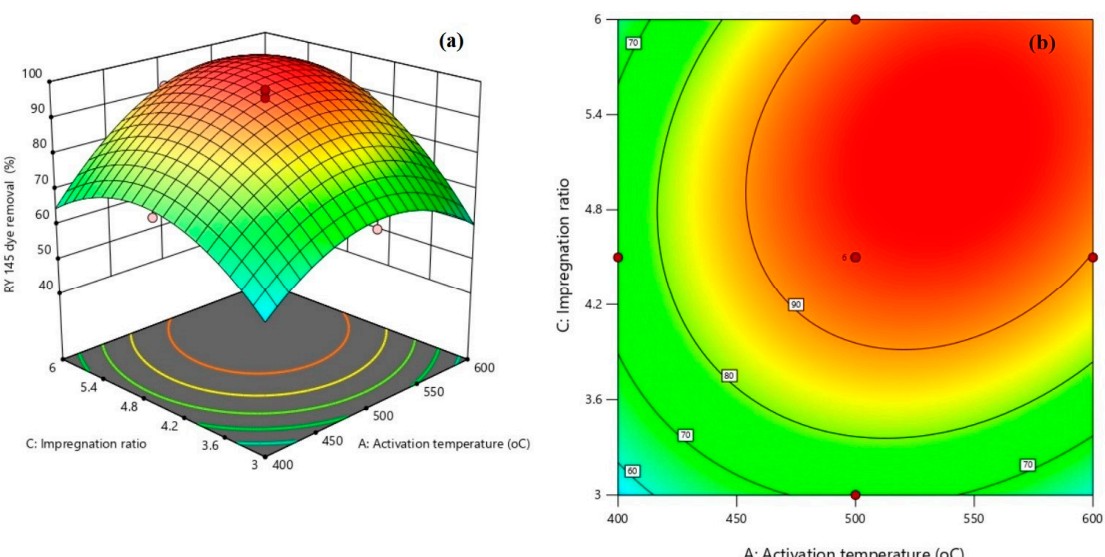

**Figure 5.** Response surface plots for the effects of $H_3PO_4$ used and activation temperature on the 3D response (**a**) and its corresponding counter plot (**b**).

### 3.2.3. Interaction Effect between $H_3PO_4$ and Heating Time

*The 3D response surface* prediction and its corresponding counter plot for the interaction effect of $H_3PO_4$ and heating time are illustrated in Figure 6a,b. The optimal value for the $H_3PO_4$ and activation time were predicted to be 5.0615 g and 1.992 h, respectively, to obtain 97.35 % of dye removal.

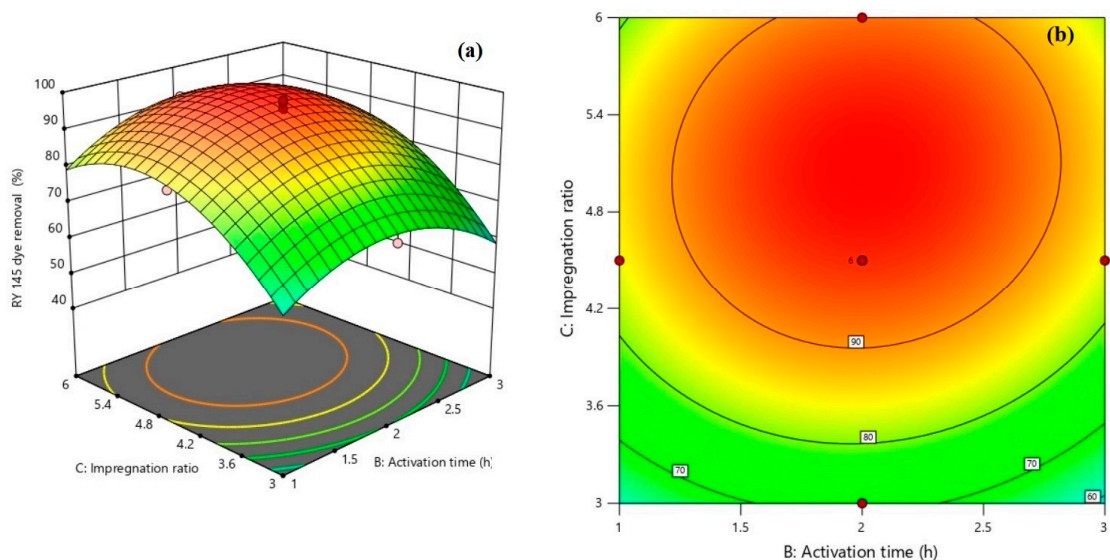

**Figure 6.** Response surface plots for the effects of $H_3PO_4$ used and heating time on the 3D response (**a**) and its corresponding counter plot (**b**).



### 3.2.4. Optimization and Model Validation

Optimization of the selected parameters for preparing the efficient TSAC was carried out using the statistical optimization technique. To determine the optimal production conditions that have the highest RY 145 dye removal efficiency, simultaneous evaluation of the factors involved in the preparation of the activated carbon (activation temperature, activation time and impregnation ratio) at various levels was performed using DOE. Table 7 shows the predicted optimal production condition by numerical statistical method. To validate the model optimal condition for TSAC production condition, experiments with the optimal condition were carried out and examined for its dye removal efficacy in triplicates. From the validating experiments, the actual dye removal efficiency was determined to be 98.53%, whereas the RSM predicted removal efficiency by the model was deduced as 99.2571%. The relative percent error for the both actual and model predicted results was 0.73%, which showed a reliability of the model along with goodness of best fit to the experimental results. The results showed that the dye removal efficiency of the obtained TSAC product was comparable to the dye removal efficiency of different activated carbons studied elsewhere by other researchers [5,35,36].

**Table 7.** Experimental confirmation of model predicted dye removal by the optimal TSAC.

| Optimum Condition | | | RY 145 Dye Removal | | |
|---|---|---|---|---|---|
| Activation Temperature (°C) | Activation Time (h) | Impregnation Ratio | Predicted Value (%) | Experimental Value (%) | Error (%) |
| 539 | 2 | 1:5 | 99.2571 | 98.53 | 0.73 |

### 3.3. Characterization of the Optimized TSAC

The activated carbon prepared using the optimized production condition exhibited a yield of 91.43%, which was calculated based on Equation 3. The TSAC preparation condition with the optimal RY 145 dye removal efficiency was determined using the RSM; the same has been confirmed using validation experiment. Further, the TSAC produced using the optimal condition was characterized by different techniques. In such a way, the Fourier transformed infrared (FTIR) analysis (Figure 7a,b) shows the functional groups on the raw TS and activated TS. It is important to investigate the presence of functional groups present in the TSAC that are responsible for the RY 145 dye adsorption. In the figure, TSAC has absorption bands at 3725 cm$^{-1}$, 3029 cm$^{-1}$, 1891 cm$^{-1}$, 1728 cm$^{-1}$, 1550 cm$^{-1}$. The bands confirm the existence of hydrate ($H_2O$) and hydroxyl (-OH), carboxylic acid and carbonyl on the TSAC. The raw TS has bands at 3332, 2918, 1737, 1614, 1357, 1248 cm$^{-1}$. These bands show the presence of hydroxyl, hydrate ($H_2O$) and hydroxyl (-OH), carboxylic acid and carbonyl compound. In [27], a peak was observed at 3421 cm$^{-1}$ on the adsorbent prepared from TS which was due to hydroxyl (O-H) groups stretching vibrations. In the raw TS and activated carbon prepared using TSAC, in [15], peaks were observed at (1700–1450 cm$^{-1}$) and 3700–3150 cm$^{-1}$) which indicated the carboxylic and carbonyl groups and -O-H stretching vibrations, respectively. The implication of the presence of the functional groups is that reactive dyes possess strong affinity to cellulose and bio-adsorbents such as hydroxyl group. Thus, the presence of hydroxyl functional group in the TSAC might be responsible for its strong capacity for adsorptive removal of the RY 145 dye [39].

The X-ray diffraction (XRD) Pattern on TSAC is shown in Figure 8. The result shows only one small peak that appears at approximately 2θ = 23.06°, which indicates the formation of crystalline structure [40,41]. Strong peaks were absent in the majority of the XRD patterns. Broad diffraction patterns that correspond to the 2θ angle were observed between 10.46° and 22.51°; 23.74° and 32.39°. Those broad peaks are the indicators of formations of amorphous structures, which is an advantage for adsorption [41]. The obtained results seem to be very close to those of a previous study on TS [27].

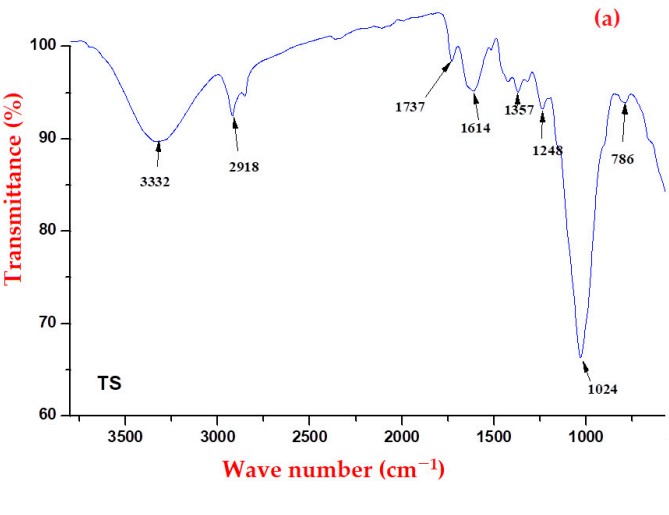

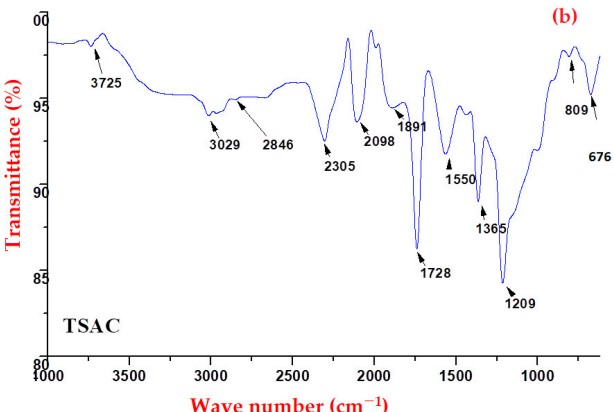

**Figure 7.** FTIR analysis results for Raw TS (**a**) and TSAC (**b**).

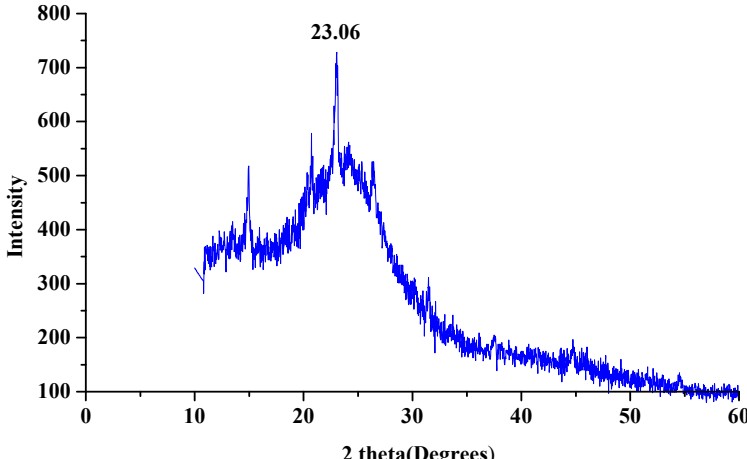

**Figure 8.** XRD patterns of the TSAC.

In order to investigate the variation in surface texture and morphology, SEM analysis was undertaken for the samples of raw TS, TSAC, and TSAC after adsorption. The analysis results are presented in Figure 9a–c, respectively. The raw TS had a relatively smooth surface. The activated carbon surface was observed to be rough and dense due to pores and active sites. The inferences confirmed the effective adsorbent formation with porous structures on its surfaces, which can be potential characteristics for dye adsorption capacity.

The *Teff* straw-activated carbon after adsorption (Figure 9c) had small pores and structures that were attributed to the adsorption of dyes on its surface and the unavailability of active sites. The surface texture and morphology of the TSAC prepared in this research was determined to be similar to the previous reports [15,26,27].

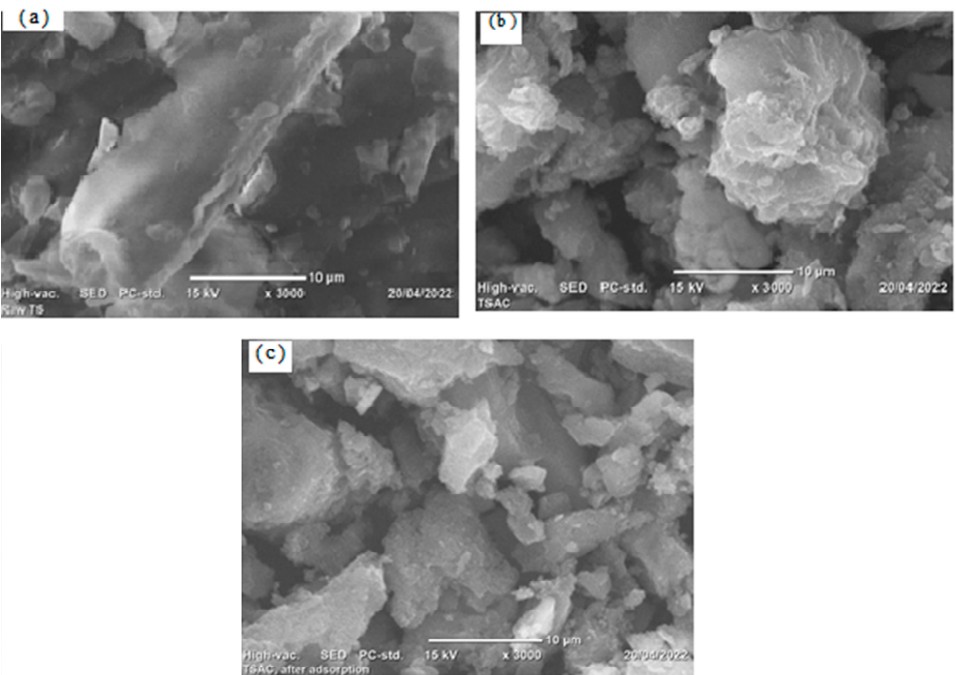

**Figure 9.** SEM images of Raw TS (**a**), TSAC before (**b**), and TSAC after (**c**) adsorption.

Aiming to determine the pH$_{pzc}$ on optimal TSAC, pH drift method was executed. First, six sets of 50 mL of 0.01 mol/L NaCl solution with initial pH values 2, 4, 6, 8, 10, and 12 were prepared in Erlenmeyer flasks (250 mL). After adjusting the initial pH (pH$_{initial}$) with 0.1 mol/L HCl and NaOH solution, in each flask, 1g of the TSAC was added and stirred at 120 rpm for 48 h. After 48 h, the pH (pH$_{final}$) was measured. The pH$_{pzc}$ of the TSAC was then determined from the graph shown in Figure 10 to be 2.1. The surface area analysis of the TSAC showed a value of 671.347 (m$^2$/g), which is comparable to the result reported in [42] which was 627.7 m$^2$/g.

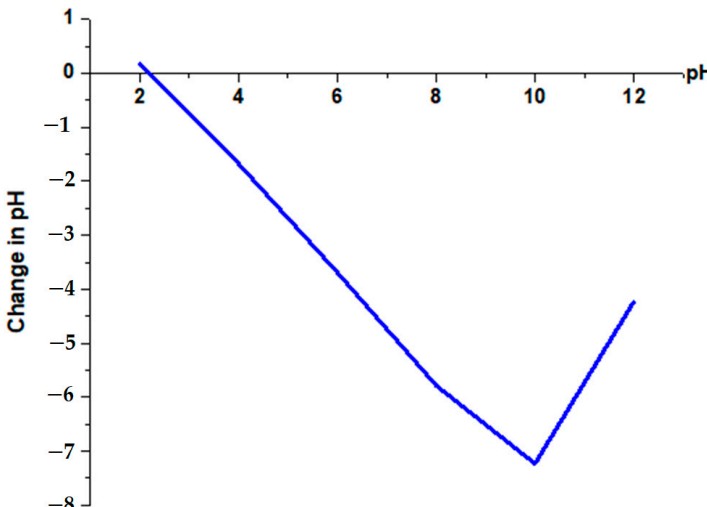

**Figure 10.** The pHpzc value of TSAC.

*3.4. Effect of Individual Process Design Parameters on the RY 145 Dye Removal Efficiency Using the Optimal TSAC*

In order to determine the optimal RY 145 dye adsorption conditions, selected process key factors were subjected to study for investigating their individual effect. Hence, the optimal TSAC was used to investigate the optimum contact time, solution pH, adsorbent dose, and initial dye concentration.

### 3.4.1. Effect of Contact Time

The effect of contact time on the removal efficiency of TSAC is shown in Figure 11. In the present study, the different contact times are used for the experimental adsorption studies, such as 30, 60, 90, 120, and 150 min. Other factors (Ph = 6, dose 0.3 g/L of solution, concentration 0.3 g/L and agitation speed of 120 rpm) were kept constant for all experiments. In the first 30 min, the percentage removal was determined to be rapid, and then followed a constant rate until it attained equilibrium condition at 120 min. This might be due to the presence of reactive sites on the adsorbent external surface which when saturated results in slower entrance rate from the exterior surface to the interior pores until equilibrium [43]. However, further increase after equilibrium resulted in a decrease in removal efficiency due to the desorption of the dye in to solution after attaining the equilibrium [44,45].

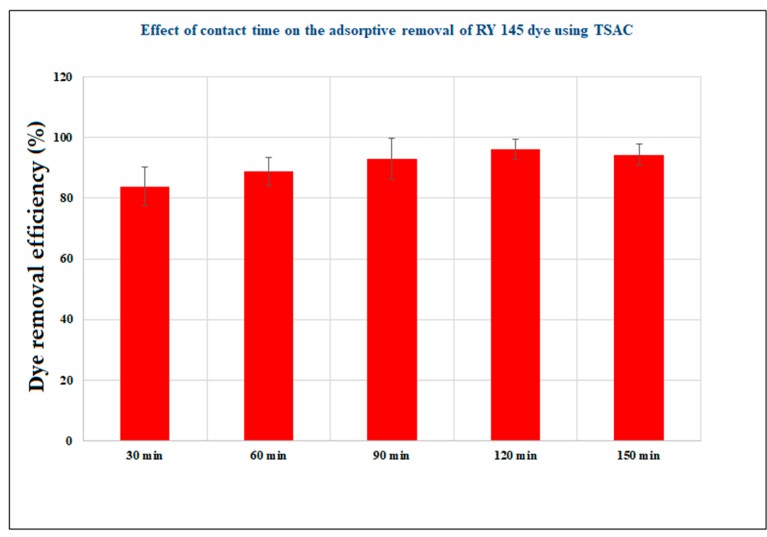

**Figure 11.** Effect of contact time on the adsorption of RY 145 dye on TSAC ([RY 145 dye] = 0.3 g L$^{-1}$; Adsorbent dose = 0.3 g/L of solution; pH = 6; shaker speed = 120 rpm).

### 3.4.2. Effect of Solution PH

Studies have proven that the solution pH can significantly affect the adsorptive uptake of adsorbate molecules. This is most probably through its effect on the adsorbent surface charge, the adsorbates degree of dissociation or ionization, and dye structure [43,44]. For this study, the effect of pH was investigated under the pH values of 2, 4, 6, 8, & 10. As seen in Figure 12, results showed that the dye removal efficiency decreased from 99% to 80% while the solution pH was increased from 2 to 10. It was determined that the maximum adsorption occurred in acidic region at pH value 2. However, value of pKa may influence the adsorption based on the strong/weak acid which is used to maintain the required pH to maximize the adsorptive removal. At low pH, due to protonation, the adsorbent surface charge becomes positively charged. This results in higher removal of negatively charged adsorbates. However, at high pH, due to deprotonation, both of the adsorbent and adsorbate become negatively charged which results in decrease in removal efficiency of the dye due to repulsion between them [10,45]. The inference was supported by different researchers; in this line, the maximum removal of reactive dye at low pH was reported

in [46], where the authors worked on the removal of reactive yellow 145 (RY 145) dye using bio-char derived from Groundnut Shell. In [26], highest percentage of chromium (VI) removal from contaminated aqueous solution at pH 2 was reported.

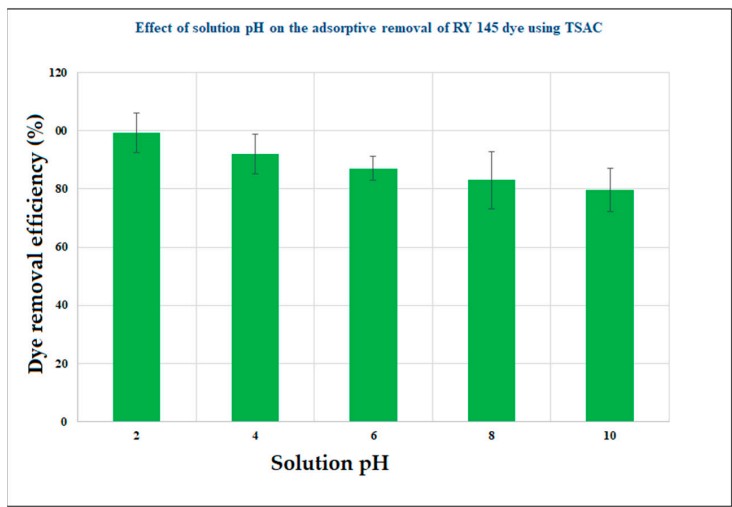

**Figure 12.** Effect of solution pH on the removal of RY 145 dye ([RY 145 dye] = 0.3 g/L dye; Adsorbent dose = 0.3 g/L of solution; contact time = 120 min).

### 3.4.3. Effect of TSAC Dose

Batch experiments were carried out at various doses of the adsorbent (0.1, 0.2, 0.3, 0.4 and 0.5 g/L of solution) while keeping the other variables constant. As shown in Figure 13, with the increase from 0.1 to 0.4 g/L of solution dose of TSAC, increase in the RY 145 dye removal was observed with the maximum removal at 0.4 g/L of solution dose. With further increase in the adsorbent dose, no significant change in removal of the dye was determined. Further increase in adsorbent dose resulted in increase of the available adsorbent sites [47]. After this level of dose, the increment of dosage did not show any significant change in the percentage removal and the percentage of dye removal slightly decreased. This may be because of the formation of aggregates and overlapping by the particles of adsorbent in the solution [48].

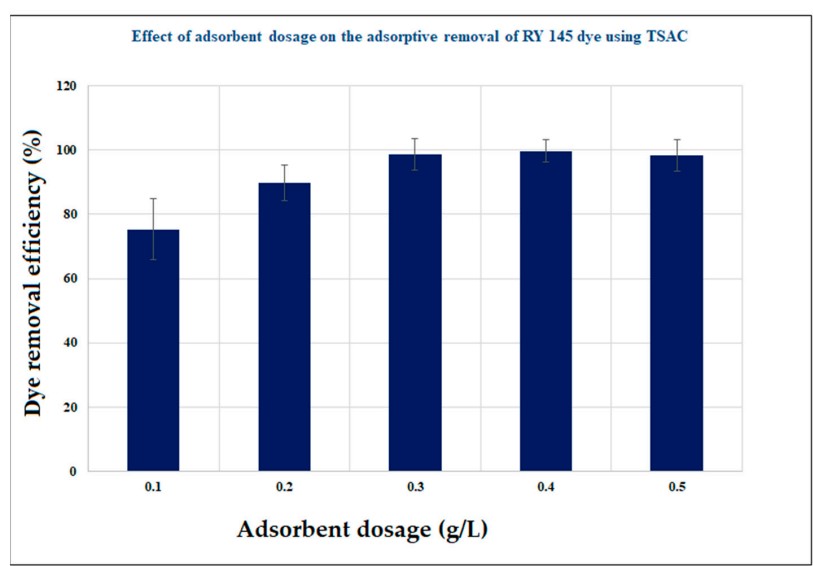

**Figure 13.** Effect of adsorbent dose on the adsorption of RY 145 dye on TSAC. ([RY 145 dye] = 0.3 g/L; Ph = 6; time = 120 min).

### 3.4.4. Effect of Initial RY 145 Dye Concentration

Batch adsorption experiments involving different initial concentration of RY 145 dye, i.e., 0.1, 0.2, 0.3, and 0.4 and 0.5 g/L were studied by keeping the other variables constant with contact time of 120 min, adsorbent dose 0.3 g/L of solution and pH 6. As can be seen in Figure 14 below, for a fixed dose of the adsorbent, with an increase in initial dye concentration, percentage of RY 145 dye removal also increased. The reason may be due to the increase in concentration gradient between the aqueous solution and the solid phase [49]. Since the adsorbent dose was fixed, there were a fixed number of active sites; as a result, percentage removal decreased with increasing concentration after 0.3 g/L RY 145 dye concentration. With further increase in concentration of the dye after gaining optimum level, the removal efficiency was determined to gradually decrease. It is possible that at the beginning, the adsorption rate increases with increase in initial concentration of adsorbate due to more availability of binding sites [50]. At a low initial dye concentration, the dye solution was very dilute, the dye solution had a high concentration gradient, and a relatively low sorption occurred [51,52].

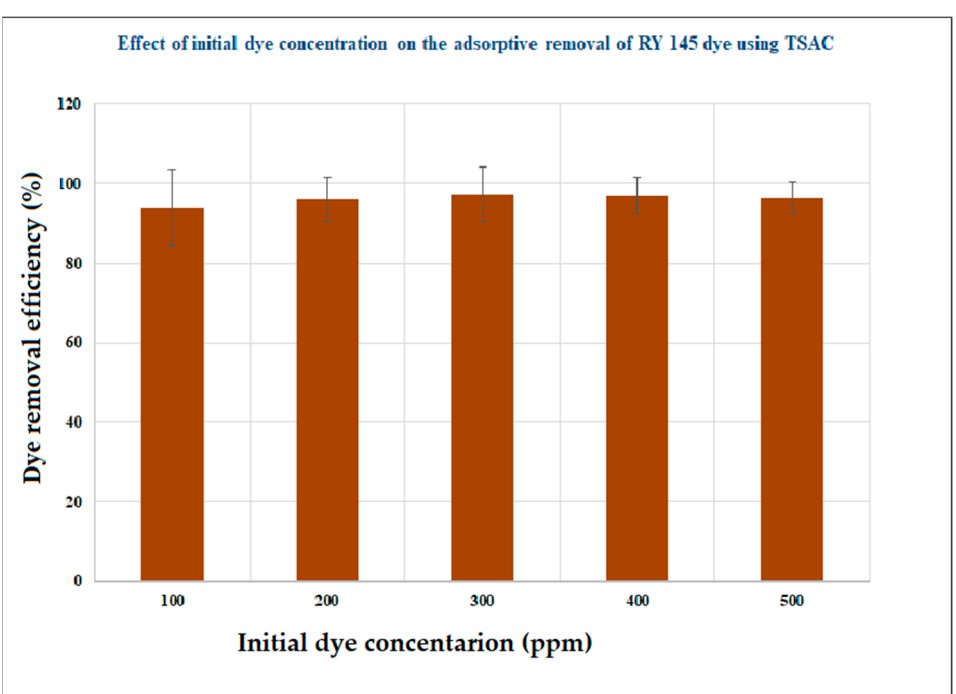

**Figure 14.** Effect of initial concentration of RY 145 dye on its adsorption onto the TSAC (Adsorbent dose = 0.3 g/L of solution; pH = 6; time = 120 min).

### 3.5. Removal of COD from Real Textile Effluent Using the Optimal TSAC

Batch experiments were conducted in triplicate to determine the efficiency of the TSAC in removing COD from real textile effluent. As shown in Table 2, the initial COD concentration of the real textile effluent was 804 mg/L. The optimized TSAC removed 76% of the COD and reduced it to an equilibrium COD concentration of 192.96 mg/L, which is below the Ethiopian EPA COD release standard that is 250 mg/L. The result indicated that TSAC could be used as an alternative for COD removal from textile effluent (Figure 15). The COD removal efficiency of the TSAC prepared in this study is comparable to the activated carbons reported in the literature [35,48,53].

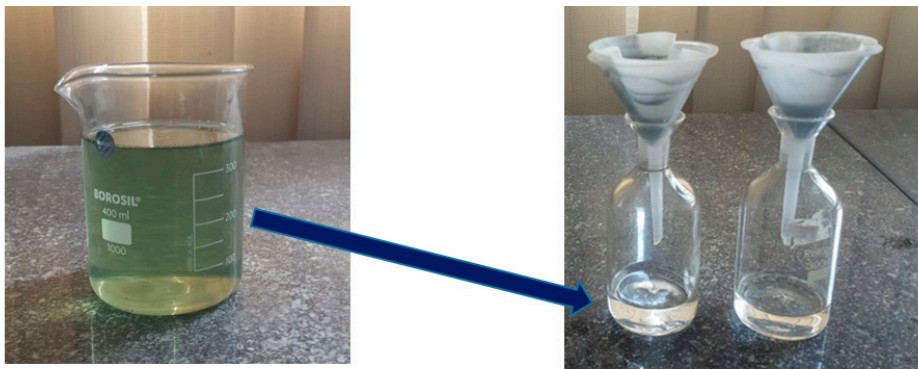

Textile effluent before treatment                     Textile effluent after treatment

**Figure 15.** The images of the textile waste effluent before and after treatment.

## 4. Conclusions

In this study, *Teff* straw (TS), one of the under-looked feedstock for bio-based activated carbon, was examined to employ as a low-cost bio-adsorbent to remove the toxic contaminates, specifically from textile wastewaters. The adsorptive removal of RY 145 dye was subject to examination. Since the preparation condition can crucially influence the performance of bio-adsorbent, selective preparation factors such as activation time, activation temperature, and impregnation ratio were optimized. Further, they were determined to be 2 h, 539 °C and 1 g of $H_3PO_4$ to 5 g TS, respectively, through RSM. Using the TSAC prepared, optimal condition was characterized through XRD, SEM and FTIR. The results confirmed that there were significant active sites and functional groups that could be comparable with properties of the activated carbon reported elsewhere. The TSAC showed an excellent result by removing 98.53% of RY 145 dye from the aqueous solution. Further, the potential of TSAC was examined for determination of the individual effect against contact time, solution pH, adsorbent dosage, and initial dye concentration. In addition, the TSAC was investigated on its removal performance of COD in real textile waste effluent. The results confirmed that 76% of COD was removed from the real textile waste effluent that was an appreciable outcome to meet Ethiopian Environmental Protection Authority (EPA) standard. Aiming towards an identification of economically viable and environmentally friendly detoxification process for textile effluent, such investigation provides a robustious idea for dye removal techniques. Continuing investigations will be focused on the applications such as dye removal from industrial waste effluents in fixed-bed column system as well as regeneration techniques for dye-laden adsorbent.

**Author Contributions:** Investigation, M.K.; supervision, E.A. and B.L.; writing—original draft, M.K.; writing—review and editing, M.K., E.A., S.V.P., Z.W., J.F. and B.L. All authors have read and agreed to the published version of the manuscript.

**Funding:** No external funding received for the research.

**Institutional Review Board Statement:** Not applicable.

**Informed Consent Statement:** Not applicable.

**Data Availability Statement:** The data used in this research are incorporated in the article.

**Acknowledgments:** The authors gratefully acknowledge the Addis Ababa Science and Technology University for the financial support. The Environmental Engineering Department of the University, for the laboratory services, and Yirgalem Addis Textile Factory are also thanked for supplying dyes and textile waste effluent for the research work.

**Conflicts of Interest:** The authors declare that they have no conflict of interest.

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
