# Peer review of "Adsorptive Removal of Reactive Yellow 145 Dye from Textile Industry Effluent Using Teff Straw Activated Carbon: Optimization Using Central Composite Design"

_water, doi:10.3390/w15071281_

Round 1
Reviewer 1 Report
The manuscript by Kifetew et al. is a thorough investigation of the effect of phosphoric acid treatment of Teff straw on its dye removal and chemical oxygen demand. The study analyzes the dye removal of activated carbon based on its experimental preparation conditions (activation time, temperature, and impregnation ratio). The “optimum” (activated carbon with maximum adsorption of Reactive Yellow 145) was further investigated. The activation carbon was characterized using FTIR, SEM, XRD, BET, and point of zero charge. Overall, the authors conclude activated Teff straw to be a suitable, low cost biosorbent of Reactive Yellow 145 from industrial wastewaters. The authors propose scientifically and industrially useful follow up studies on fixed-bed columns and regeneration techniques. The research addresses a clear need to clean up huge volumes of wastewater containing a variety of chemicals.
The manuscript is probably publishable but there are several aspects before I can recommend publication.
The authors do not clearly define what is meant by “TSAC at optimal condition.” Adsorption time, reaction time, oven temperature, activated carbon yield, cost, etc. are all factors. Optimization could mean cost, time, adsorption, etc. What is optimized should be understood from the first usage.
Line 120: define λmax
Section 2.4.1 - Equation 2.2 and its variables should be checked. The subscripts, superscripts, and multiplication are either incorrect or not rendering properly.
Section 3.4.2 -Was the pKa measured? The explanation of the effect of pH and protonation state discussion could be simplified by referring to the pKa of the charge groups.
Teff straw is a waste product of a local food stable. Would adoption of the straw as a biosorbent influence the food supply for humans or livestock?
The conclusions may be overly generalized to other dyes. Could the authors comment?
The conclusion section primarily summarizes the experimental conditions and leaves out some of the interesting findings of the work. For example, the finding that “activation temperature had significant effect on the dye removal, whereas activation time had less effect” is advantageous for industry and other wastewater treatment set ups but it is not mentioned in the conclusion. The authors could increase the impact of their research by strengthening their conclusions.
There are other minor comments:
COD is not defined in the abstract.
The abbreviation CCM is defined but not used. Alternatively, it has also been defined as CCD. Reconsider usage.
Line 57. “When living organisms are exposed to these chemicals, result in allergy, cancer, eye or skin infections, dermatitis, vomiting, heart problem, and so forth [5]” The authors draw a direct line between chemical exposure and health problems. Is it more accurate to state that exposure may result in or significantly increases risk of health issues?
There are several typos throughout. For example,
Section 2.2: “uses widely” à “is used widely”
Section 2.3: Typo “Removaing”
What is “oT” of the activation temperature on lines 221?
The horizontal and vertical axes of Figure 3 should have appropriate labels and units. What is the meaning of the color of the data points?
Legend is unnecessary in Figures 10-14.
Caption of Figure 12: pH of what? The wastewater effulent?
References: The journal names should be abbreviated and capitalized throughout.
Reviewer 2 Report
The article water-2270579 presented the preparation and optimization of an adsorbent to remove a dye and treat a real textile effluent sample. However, several corrections are necessary.
Line 36: Change “dyes from wastewater” to “pollutants from textile wastewater”;
It is important to cite recent review articles on the treatment of textile effluents: https://doi.org/10.1016/j.jwpe.2021.102273; https://doi.org/10.1016/j.cscee.2022.100230;
Lines 54-57: It is inappropriate describing that COD is an ingredient of textile wastewater. Rephrase this sentence;
Line 84: Change “treat” to “remove”;
Line 91: It is recommended to justify the color chosen at the end of the introduction;
Lines 102, 398, 399: change “M” to “mol/L”;
Line 110: until the pH value equals 7? In Figure 1 it is necessary to correct for pH = 7. The reaction medium can be neutral, acidic or basic; pH can be equal to, above or below 7. Only values ​​are assigned to pH;
Line 117: This sentence is wrong;
Line 120: lmax for many dyes is dependent on the pH of the solution. Did the authors assess the effect of pH on lmax for this dye? It is necessary to use an analytical curve for each pH tested;
Lines 119-121: delete this sentence, as this information is repeated in table 1;
Line 119: Although COD is more important, authors should also present color removal data for the treated effluent;
Lines 149-157: It is necessary to mention the pH value in the optimization tests;
Line 351: correct to “TSAC”;
Lines 358-368, 402-404: Rephrase these sentences, as they are very badly written;
Line 426: Throughout the text, the amount of adsorbent must be reported in g/L of solution;
Line 440: correct to “biochar”;
Figures 11-14: If these experiments were not carried out in triplicate, this will compromise the discussion of the results, as without error bars the results are unreliable;
Line 477: The authors mentioned that the novelty of this work would be the use of the prepared adsorbent to treat an effluent, but the amount of data obtained in the treatment of the real effluent is minimal. Authors should at least show images of the effluent before and after treatment.
Round 2
Reviewer 2 Report
The article water-2270579 showed improvements in this second version, but other corrections are needed.
Comments:
- Lines 19, 101, 513: “potency” or “potential”?
- Line 21: correct to “temperature”;
- Line 23: correct to “model”;
- Line 92-94: rephrase the sentence to “Reactive Yellow 145 dye is one of the azo dyes that are widely used at large-scale as coloring agents in the textile industry”. A reference to this and the following sentence must be included in the text;
- Line 124: In Figure 1 it is necessary to correct for pH = 7. The reaction medium can be neutral, acidic or basic; pH can be equal to, above or below 7. Only values ​​are assigned to pH;
- Lines 132-134: delete this sentence, since such aspects are cited in table 1;
- Line 204: correct to “process”;
- As the experiments shown in figures 11-14 were performed in duplicate according to author`s reply, this information must be mentioned in the text. In addition, error bars must be inserted in these same figures.
